# Lineage tracing reveals fate bias and transcriptional memory in human B cells

Michael Swift[1] , Felix Horns[2], Stephen R Quake[3,4,5]

We combined single-cell transcriptomics and lineage tracing to understand fate choice in human B cells. Using the antibody sequences of B cells, we tracked clones during in vitro differentiation. Clonal analysis revealed a subset of IgM+ B cells which were more proliferative than other B-cell types. Whereas the population of B cells adopted diverse states during differentiation, clones had a restricted set of fates available to them; there were two times more single-fate clones than expected given population-level cell-type diversity. This implicated a molecular memory of initial cell states that was propagated through differentiation. We then identified the genes which had strongest coherence within clones. These genes significantly overlapped known B-cell fate determination programs, suggesting the genes which determine cell identity are most robustly controlled on a clonal level. Persistent clonal identities were also observed in human plasma cells from bone marrow, indicating that these transcriptional programs maintain long-term cell identities in vivo. Thus, we show how cell-intrinsic fate bias influences differentiation outcomes in vitro and in vivo.

## Introduction

A key focus of developmental biology is the relationship between the molecular milieu of a progenitor cell and its differentiation outcomes. These outcomes are variously referred to as cell fate, cell identity, or cell state. Lineage tracing offers a powerful way to map which progenitor cells adopt which cell fates. Even rudimentary cell labeling techniques show clonally related offspring are biased toward similar cell fates (Whitman, 1878), and recent technological advances confirm the same with greater throughput and resolution. However, the contribution of cell-extrinsic versus cell-intrinsic molecular factors as determinants of cell fates remains largely uncharacterized.

To better understand cell fate determination, multiple groups have used high-throughput sequencing to measure endogenous or transgenic DNA barcodes as labels of cellular lineage (Lu et al, 2011; Naik et al, 2013). Recently, it is possible to use high-throughput sequencing to perform both lineage tracing and transcriptomics in single cells. This combination directly measures the molecular relationships between progenitors and their offspring, allowing stronger inference of molecular determinants of cell fates (Biddy et al, 2018; Ludwig et al, 2019; Weinreb et al, 2020). For example, by analyzing the transcriptomes of lineages biased towards efficient reprogramming outcomes, Biddy et al were able to identify a previously uncharacterized gene which increased stem-cell reprogramming efficiency by threefold.

In the human immune system, clonal lineages of leukocytes rapidly proliferate while adopting diverse cell fates. This dynamic occurs in vivo as a response to varied pathogenic challenges such as viruses, bacteria, or cancer. Spatially organized cellular structures, called germinal centers, orchestrate this process in vivo. However, in vitro differentiation protocols using only T or B cells can recapitulate important features of the germinal center, and provide valuable insight into the process (Deenick et al, 1999). In vitro, a researcher can control most extrinsic factors, such as cell density, cytokine cocktails, and media compositions, allowing them to study cell-intrinsic differentiation programs. In the case of B and T cells, well-controlled extrinsic conditions still reliably generate a large diversity of cell fates, indicating a strong contribution of intrinsic cell diversity to population-level diversity seen in vivo (Cheon et al, 2021). The question of how gene expression responds to extracellular stimuli, while a cell maintains its identity, is generally poorly understood. And the transcriptional programs underlying cell-intrinsic clonal fate bias remain largely uncharacterized. Furthermore, how extrinsic signals, intrinsic state, and clonal population structure interact to determine the dynamics of the B-cell immune response remains poorly understood.

Here, we gain insight into some of these gaps in knowledge by obtaining lineage and single-cell RNA-sequencing measurements of differentiating human B cells. We used the B-cell receptor (BCR) gene as a genetic lineage marker and we paired this information

[1]Department of Chemical and Systems Biology, Stanford University, Stanford, CA, USA    [2]Division of Biology and Bioengineering, California Institute of Technology, Pasadena, CA, USA    [3]Department of Bioengineering, Stanford University, Stanford, CA, USA    [4]Department of Applied Physics, Stanford University, Stanford, CA, USA    [5]Chan Zuckerberg Biohub, San Francisco, CA, USA

Correspondence: michael.swift@stanford.edu; steve@quake-lab.org

with a transcriptomic readout of cellular identity during an in vitro differentiation of human B cells. These multimodal data allow us to quantitatively infer the intrinsic biases B cells have towards specific cell fates, analyze the clonal dynamics of in vitro B-cell activation, and characterize transcriptional memory both in vitro and in vivo.

# Results

### In vitro–differentiated B cells recapitulate major aspects of in vivo B-cell development

To study differentiation and fate choice of human B cells, we performed an in vitro differentiation protocol using primary human B cells from healthy donors. We simulated T cell–dependent activation of B cells using a cocktail of cytokines (CD40L, IL2, IL4, IL10, and IL21); (see the Methods note 1 section). These cytokines induced proliferation, class-switch recombination (CSR), and reprogramming into terminally differentiated Plasma cells. We performed single-cell RNA-sequencing on time course samples from this protocol (Fig 1A), which furnished us with population-genetic and transcriptional information about B-cell differentiation. In addition, we contextualized our in vitro differentiation by (1) integrating our single-cell RNA-sequencing data with publicly available data from 10X Genomics and (2) measuring CD138+ bone marrow plasma cells from a separate donor (i.e., terminally differentiated B cells in vivo). After quality control and bioinformatic exclusion of non-B cells, we obtained the mRNA transcriptome and VDJ sequences for 29,703 B cells from our six samples (Fig 1B and C).

Dimensionality reduction by principal component analysis and UMAP (McInnes et al, 2018) of the single-cell transcriptomes revealed several distinct cell states. We automatically annotated cell states with CellTypist (Figs 1D and S1B) (Domínguez Conde et al, 2022), and more granularly based on multimodal information we collected about the BCR (Fig S1C). We quantified the relative abundances of these algorithmically determined cell states over time, and found dramatic changes (Fig 1E). First, we noted that non-B cells present in the input rapidly became undetectable by day 4 (Fig S1D), which shows the specificity of the cytokines for B-cell expansion. Other notable dynamics included a threefold decrease in the relative abundance of plasma cells from Day 0 to 4 and a 1,000-fold increase in Proliferative Germinal center B cells. Finally, we observed substantial amounts of cell death (30%) by Day 4, implicating cell death as a major contributor to the population dynamics (Fig S1E).

### Measuring VDJ mutation status allows inference of population-level cell-fate biases

Upon experiencing antigenic stimulation, naive B cells genetically diversify their BCR by accruing mutations in their germline VDJ genes (somatic hypermutation), as well as switching expression of constant region genes through DNA deletion events (CSR). These endogenous processes have been used to make lineage inferences in vivo (Horns et al, 2016). We reasoned we could use these endogenous genetic alterations in the BCR as estimators of the initial cell state of any given cell detected in vitro. For example, we infer a B cell with an unmutated BCR detected in the time course likely arose from a naive B-cell progenitor.

We assessed the validity of this approach by quantifying the concordance between transcriptionally defined memory and naive B-cell states and categorically delineated somatic hypermutation levels (germline, mutated, heavily mutated). In general, the concordance between mutational and transcriptionally defined cell state categories was high (Fig S2A–C). For example, in the Day 0 population, more than 97% naive B cells possessed an unmutated BCR gene, also known as a germline gene, and less than 11% of plasma cells were classified germline. There was also a continuum of transcriptional states which we labeled "B cells," which encompasses a gradient of cell identities between switched-memory B cells and naive B cells. Critically, we observed no appreciable evidence of hypermutation in our in vitro conditions (Fig S2D), consistent with prior literature showing BCR stimulus is necessary for in vitro hypermutation (Bergthorsdottir et al, 2001).

Given the germline and mutated categories classified naive and naive transcriptional states in the input population with high specificity, we used the hypermutation level measured for in vitro differentiated B cells as a confident prediction of whether their progenitor cell was a naive or naive B cell. We found the progeny of hypermutated B cells increased twofold in relative abundance over the course of the culture, showing hypermutated B cells are intrinsically twofold more persistent in vitro, in our conditions (Fig 2A). This is consistent with orthogonal measurements, which report memory B cells are on average one division ahead of naive B cells when cultured in vitro (Tangye et al, 2003).

We next analyzed how the mutation status was related to their transcriptional identity (Fig 2B). On Day 0, we found what we expected in the peripheral blood. For example, transcriptionally identified plasma cells were four times more often mutated than germline. However, by Day 4, germline cells populated the plasmablast/cell state almost as often as mutated cells, definitively linking naive B-cell progenitors to plasmablast phenotypes in this culture system, showing that circulating naive B cells can rapidly adopt a plasmablast-like phenotype. As the differentiation proceeded, mutated B cells began to repopulate the plasmablast/cell compartment, suggesting most naive B cell–derived antigen-secreting cells are short-lived.

We continued to use the mutation status of the IgH locus as a lineage marker, which allowed us to understand the population dynamics of cell states in vitro. To better understand differences in the early activation programs of mutated and germline B cells, we analyzed the differentially expressed genes (DEGs) between these subsets of activated B cells (i.e., mutated versus germline Proliferative Germinal Center B cells). We found mutated B cells were likely to express genes involved in T-cell interaction such as CD70, CCL17, and CCL21 (Fig 2C). It is known that memory B cells are intrinsically licensed to enter an inflammatory state which activates T cells (Liu et al, 1995; Good et al, 2009), and our results describe the gene expression program which orchestrates this propensity. In contrast, germline B cells were biased toward expressing SELL, CLEC2B, and proliferative markers, suggesting naive B cells are intrinsically primed to home into the lymphatic system and proliferate in germinal centers.

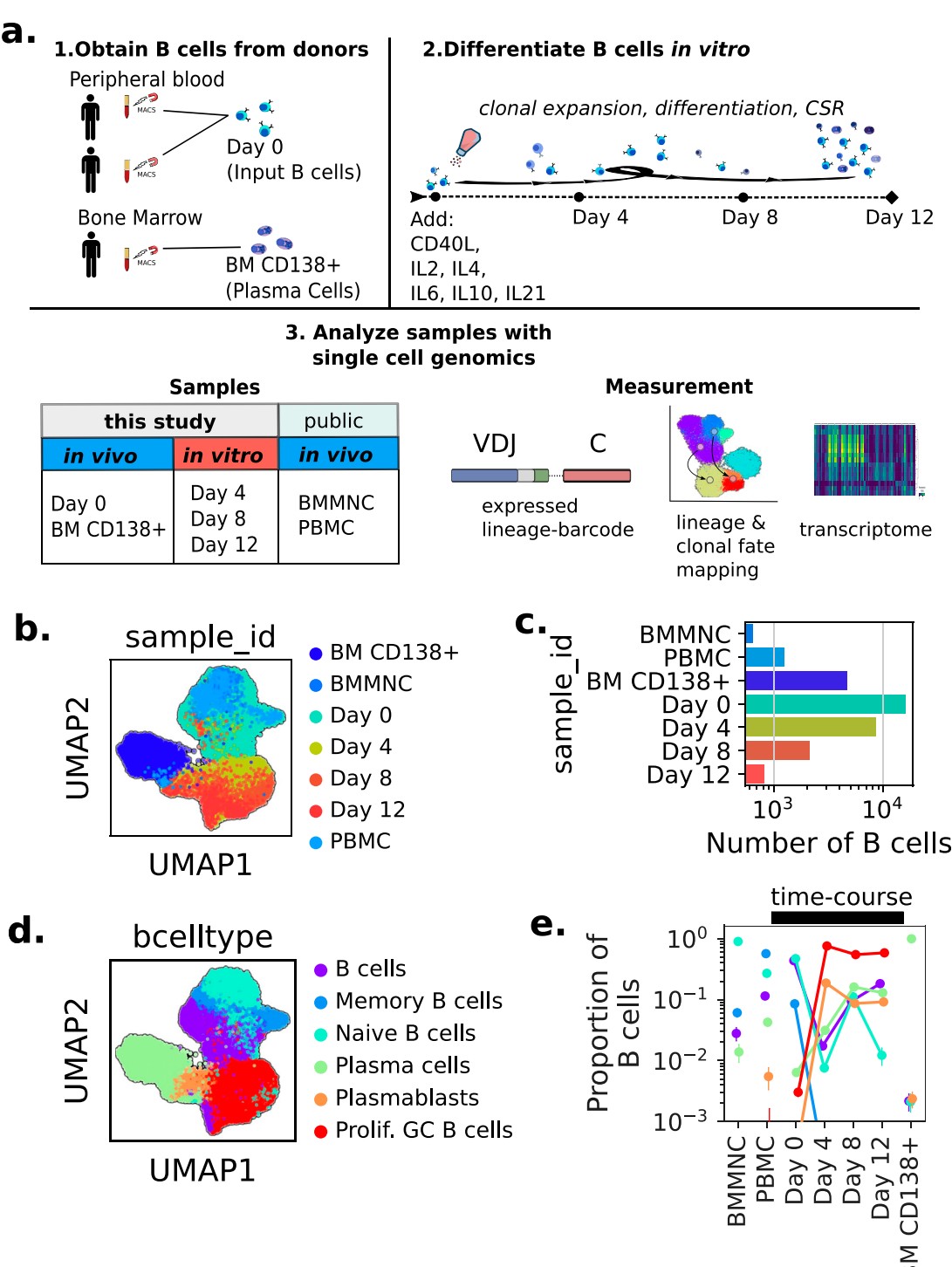

**Figure 1. Experimental overview for studying in vitro B-cell dynamics using integrated single-cell genomics and lineage tracing.**
**(A)** Experimental overview. (1) B cells or Plasma cells were purified from blood or bone marrow, respectively, using MACS. (2) B cells from the same purification were stimulated with the StemCell B cell Expansion kit. Samples of the in vitro differentiation were collected on days 4, 8, and 12. (3) Schematic of the single-cell genomic data collected and analyzed. **(B)** Principal component analysis and UMAP embedding separates cells into distinct clusters. Each dot is a cell, colored by sample of origin. **(B, C)** Countplot of the numbers of B cells passing QC for each sample (colors same as (B)). **(D)** UMAP embedding with cell type annotations; see Fig S1F for cell type–defining genes. **(E)** Proportion of B-cell types in each sample. Samples from the time course are connected. All error bars are 95% confidence intervals calculated by resampling.

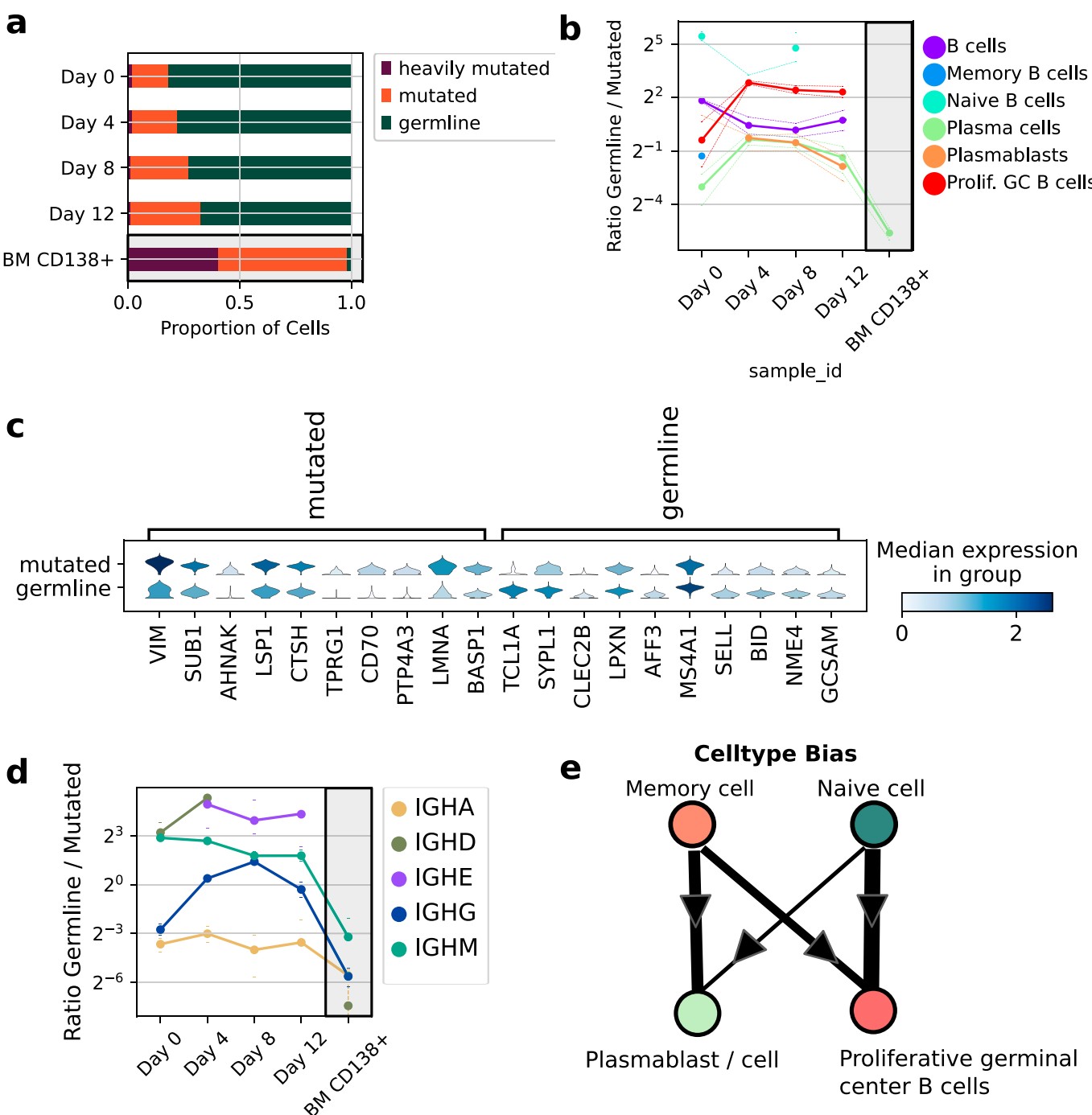

**Figure 2. Characterization of cell-intrinsic phenotypes using VDJ mutation status.**
**(A)** Proportion of germline, mutated, and heavily mutated B cells in the samples; gray boxes illustrate BM CD138+ which is not part of the time course but serves as a comparison sample. **(B)** Ratio of germline to mutated cells in each cell state over the time course. **(C)** Top differentially expressed genes in between mutation categories in Proliferative germinal center B cells. Gene expression is log base 2 umis per 10,000. **(D)** Ratio of germline to mutated cells for each isotype group. Gray box as in (A). **(B, E)** Illustration of the inferred cell-type biases from (B). Error bars in all figures are 95% confidence intervals calculated by resampling.

Our measurement of phenotypes was not limited to the transcriptome because B cells generated additional phenotypic diversity in vitro through CSR. Generally, as naive IGHM+ B cells experience cytokinetic/antigenic stimulation, they class-switch to any of the IGHA, IGHG, or IGHE genes. This process diversifies the immune response by producing antibodies with the same specificity, but different effector functions. We quantified the in vitro dynamics of CSR through the lens of mutation status, which revealed strongly different fate biases between germline and mutated cells (Fig 2D). Most strikingly, B cells which switched to

IGHE were almost exclusively derived from germline progenitors: the ratio of germline IGHE cells to mutated IGHE cells was (eightfold - inf, 95% CI). In Fig 2E, we illustrate models of cell state biases which were calculated from our population lineage tracing.

## Clonal analysis reveals clonal fate bias, MZ-like B cells, and a map of class-switch events

We next used the full BCR sequence to identify clones in our dataset. Clones were defined as having identical CDR3s and using the same heavy-chain V gene. Among the 11,333 differentiating B cells, 1,911 were clonally related to at least one other detected cell (Figs 3A and S3A). Using the paired clonal and transcriptomic information, we determined that clones had very strong fate biases (Fig 3B). In this analysis, we defined fates by Leiden clustering (Fig S3B). Clones were twofold more likely to be found in a single fate, than expected given the large diversity of transcriptional states in the population and controlling for variation such as mutation status or sequencing batch. Only about 5% of clones adopted more than two fates, showing that although multi-potency was possible, strong fate biases within clones were the norm.

We detected 73 clones with family members detected at Day 0 and at a later time point (Fig S3C). We called these persistent clones. Among persistent clones, IGHM+ B cells with mutated VDJs (B cells_mutated_IGHM) was the most common progenitor state at Day 0. This suggests B cells with this phenotype are hyperproliferative compared with other B cells, as has been observed by others (Seifert et al, 2015). To test this hypothesis on our data, we modeled a scenario in which persistence was equal amongst all Day 0 clones: division rates were the same, and death rates were zero. We detected ~2X more progeny of this cell state than would be expected in the case of the aforementioned even-expansion model (Fig 3C). We wondered whether these persistent clones had a more granular identity, which was not detected by the standard single-cell clustering and differential expression approaches used to annotate them in the first place. To this end, we performed differential expression analysis on the persistent clones versus non-persistent clones found within the "B cells_mutated_IGHM" subset. This analysis revealed persistent clones were characterized by high expression of CD1C, FTX, and LPP. These clones also had low expression of TAX1BP1 and CD27 (Fig 3D). We describe these cells as MZ-like B cells because their phenotype resembles circulating splenic marginal zone B cells, which have also been shown to respond rapidly to immune challenges (Weller et al, 2004). These cells also bear a resemblance to age-associated B cells (ABCs) (Cancro, 2020).

We also used the clonal information to understand the in vitro dynamics of CSR. On the population level, we observed an order of magnitude increase in the amount of class-switched cells above the input (Fig S3C). We used the observed intraclonal isotype counts to derive a map of class-switch outcomes in vitro (Figs 3E and S3E). For comparison, we calculated the naive probability of detecting a switch given the proportions of isotype usage in the general population. For same–same isotype relationships (i.e., IGHG1 IGHG1), the map of in vitro class-switching showed more than 10-fold enrichment compared with the naive probability model. This enrichment can be explained by clonal inheritance of isotype status. We also noted

a strong divergence from this model for the IGHM to IGHA1 switch. Although the naive probability model expects a large amount of IGHM to IGHA1 switches, we detected few. These data show CSR from IGHM cells did not meaningfully contribute to the abundance of IGHA+ cells in the population, as would be expected given a lack of TGF-β in the cocktail (Stavnezer & Kang, 2009). Thus, our clonal analysis definitively clarified whether IGHA+ cells in the output came from the differentiation process or through proliferation of existing IGHA+ cells. In contrast, we noted that many intraclonal class-switching events appeared to be directly from IGHM to IGHE, showing that direct switching was more probable than sequential switching in our conditions. Given this IGHE+ cells are generally not detected in the peripheral blood in vivo, there are likely efficient intrinsic or niche-based factors which limit the appearance of these IGHE+ cells in the peripheral blood during an immune response.

## Persistence of transcriptional memory varies across genetic loci

The intrinsic biases in fate outcomes that we detected must be underpinned by the persistence of transcriptional memory at individual genetic loci. In principle, some genes may exhibit faithful transmission of transcriptional state across clonal expansion, whereas other genes may not. Thus, we sought to determine how the persistence of transcriptional memory varies across the genome. We used a permutation test on the clone labels to find genes which were less variable within clones than between.

Using this test, we identified a set of 6,937 genes with $P$-values less than < 0.01. Supporting the sensitivity of our test, we identified genes known to be clonally inherited: the light chain variable genes and the light chain constant regions. These genes were not used per se to identify clones, providing clear evidence for the validity of the test. We call the genes we identified clonally coherent genes (CCGs). One way to intuitively understand these clonal effects is to observe the cascade plot of the putative CCG, for example, IGKC (Fig 4A) (Horton et al, 2018). A clear clonal structure of high expressing clones and low expressing clones is obliterated upon permuting clonal labels. Whether we detected a CCG was highly related to its expression level. For lowly expressed genes, we often could not reject the null hypothesis, likely because their measured expression levels are dominated by technical noise such as dropout (Fig 4B).

We calculated an effect size of the variability in gene expression explained by the clonal labels called the clonal index (see the Materials and Methods section). Unsurprisingly, the Ig variable genes and light chain genes had some of the strongest effects (Fig 4C). These values provide helpful calibration for how strong clonal effects are in other genes. In contrast to the light chain, where cells generally only transcribed one constant region gene, we measured robust transcription from multiple different heavy-chain constant region genes in single cells. This transcription is consistent with the so-called sterile transcription which is necessary for CSR (Lee et al, 2001). Strikingly, we found Ig heavy-chain constant region genes such as IGHE were CCGs (Fig S4A), which was surprising given the diversity of transcriptional states measured for the locus. The quantitative clonal coherence of IgH transcription suggests faithful propagation of chromatin states

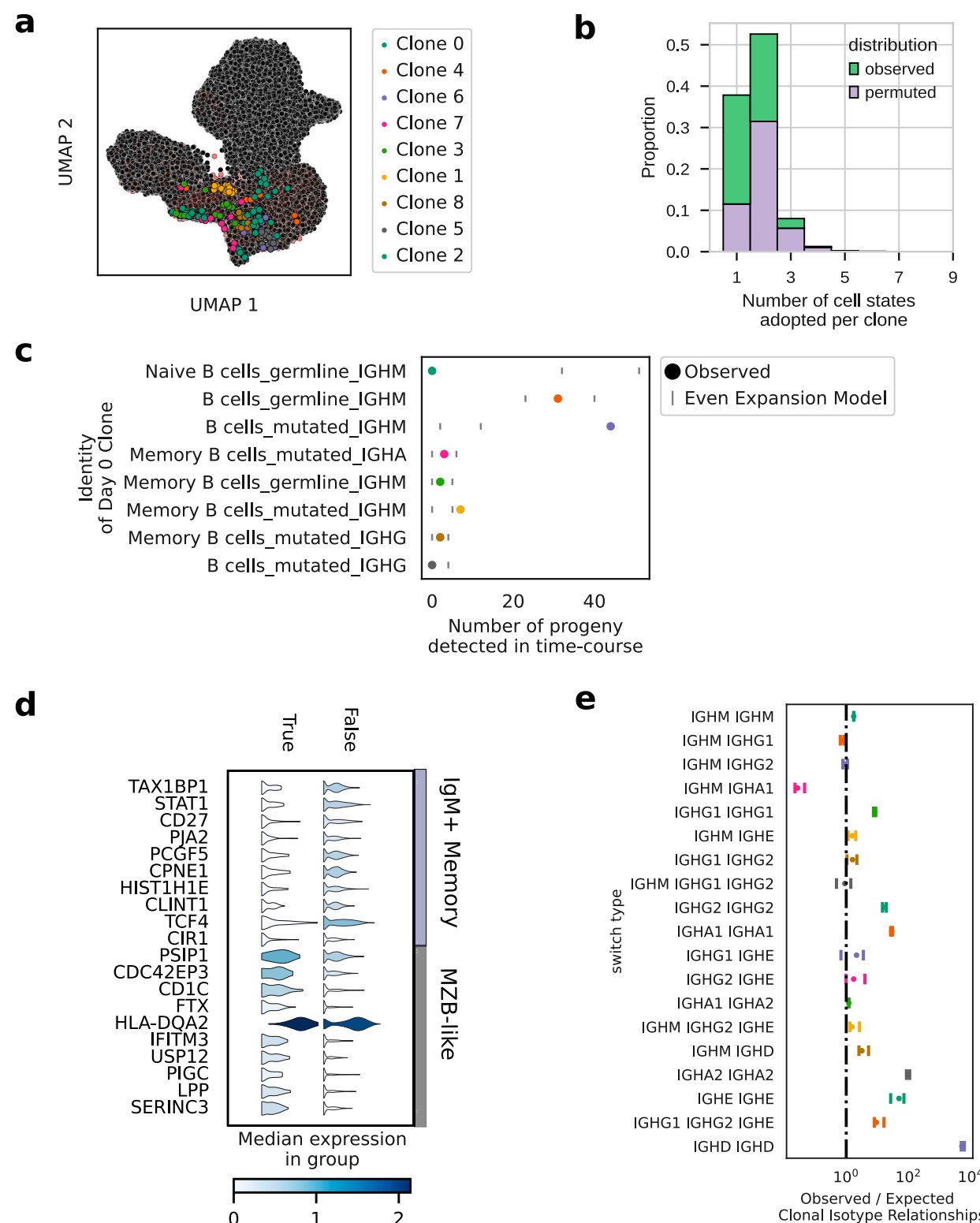

**Figure 3. Clonal families allow inference of intrinsic proliferative ability, limited clonal fate outcomes, map of class-switching in vitro.**
Error bars in all figures are 95% confidence intervals calculated by resampling. **(A)** UMAP embedding showing the largest clonal families detected. **(B)** Stacked histogram showing the number of cell fate outcomes available within clones compared with permuted clonal labels. Clone labels were restricted to permutation within their respective mutational groups to observe the effect of intrinsic bias that is not explainable by mutation status. **(C)** Progeny of the mutli-modal defined "B cells_mutated_IGHM" are overrepresented in the differentiated population. Observed data are compared with a model where the observed Day 0 clonal structure is expanded evenly for eight divisions and then randomly resampled. **(D)** Differential expression analysis of the persistent clones versus non-persistent clones in "B_cell_mutated_IGHM" population. **(E)** Ratio of observed to expected clonal isotype relationships for clonal isotype relationships that were detected.

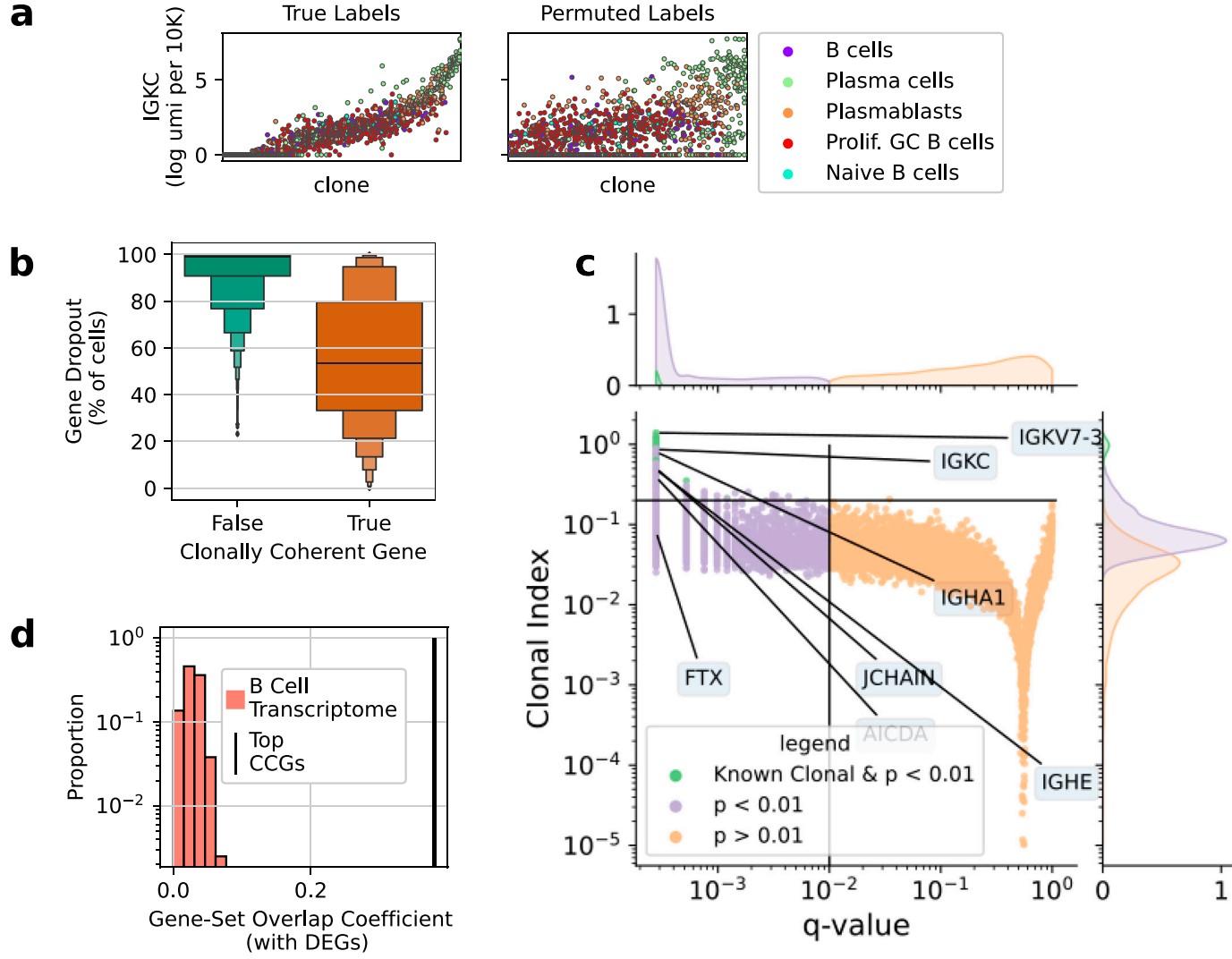

**Figure 4. Clonal transcriptional programs are strongly enriched for fate determining genes.**
**(A)** Prototypical cascade plots showing IGKC expression amongst IGKC+ clones determined via immune repertoire sequencing. Each column is a clonal family of size ≥4. Families are rank-ordered by the mean gene expression of the family. Each dot is a cell, colored by the cell state label. The true clonal families are plotted on the left and a permutation of the clonal labels is plotted on the right. **(B)** Boxen plot showing the distribution of percent dropout for all genes tested. Genes across the entire distribution of detection rates were detected as clonal genes. Genes not detected as clonal were often very lowly expressed. **(C)** A volcano plot showing results of the transcriptome-wide permutation test. Q-values are the Benjamini–Hochberg–corrected *P*-values and the clonal index is a normalized metric of expression variance described in the Materials and Methods section. Genes of interest are labeled and groups of interest are colored. Known clonal genes are the variable immunoglobulin genes. **(D)** The set of top CCGs strongly overlap with the set of top cell state defining genes compared to sets of randomly selected genes (*P* < 0.001). The null expectation is the B-cell transcriptome, which were size-matched sets of genes sampled randomly from the set of genes in at least 10% of B cells.

at the IgH locus across cell division, and may be an explanation for observations of clonal coherence in isotype usage in vivo (Horns et al, 2016).

We noticed that the CCGs with the strongest effects were often genes known B-cell fates, such as MS4A1, AIDCA, and JCHAIN. To quantify this observation, we calculated the overlap coefficient between the set of top CCGs and top DEGs between B-cell states. We observed strong agreement between the set of top DEGs and top CCGs (0.39 overlap), which was 14-fold higher than the agreement between DEGs and null-sets of genes sampled from the B-cell transcriptome (Fig 4D). Using a set of important B-cell genes curated from the literature (Morgan & Tergaonkar, 2022),

we also found an eightfold enrichment for CCGs. These enrichments show the CCGs are involved in cell fate determination, are relevant features for functional characterization, and could be used for feature selection in single-cell workflows versus highly variable genes.

Finally, we asked whether we could identify CCGs from in vivo samples. We used a cascade plot to visualize the clonal expression levels of IGKC+ bone marrow plasma cells. We observed intraclonal coherence in these samples as well, where certain clones were IGKC high, and others were IGKC low expressers (Fig S4A). When we tested the entire transcriptome we found hundreds of CCGs, which is an order of magnitude fewer genes than the in vitro data (Fig S4B).

Nonetheless, the genes we identified were important for plasma cell function such as MZB1, JCHAIN, and the IgH constant region genes. JCHAIN was detected as a CCG both in vitro and in vivo. Thus, we speculate that differentiating B-cell clones make a stable fate choice to express whether JCHAIN. Given CD38+CD138+ Plasma cells could have been generated by immune responses years in the past (Hammarlund et al, 2017), we expect we are detecting inherited transcriptional programs on the scale of years later.

# Discussion

An effective immune response requires profound and rapid differentiation while maintaining tight control of cell identity. We combined single-cell genomics and lineage tracing to investigate the differentiation process of human primary B cells. Using this multimodal measurement, we observed a large diversity of phenotypes in differentiating B cells. In the population of activated B cells, we were able to explain much of this diversity through inference of progenitor states. Populations of memory and naive B cells showed clear differences in their proliferative ability and the gene expression programs they adopted in response to the same stimulus. Importantly, typical single-cell RNA-seq measurements would not resolve this meaningful heterogeneity. At a more granular level, our clonal analysis also allowed us to characterize MZ-like B cells circulating in the blood, which again would be difficult to identify via standard single-cell RNA-sequencing techniques because of their rarity and subtle phenotypic differences on purely transcriptional level. Interestingly, these cells had very low expression of TAX1BP1. This low expression may be a molecular explanation for their high proliferation rates and proclivity towards plasma cell differentiation, as has been seen in TAX1BP1 knockout cell lines (Matsushita et al, 2016).

We found that B-cell clones were likely to choose a single-cell fate, even in the midst of the many possible fates given the diverse cytokine stimulus. This fact holistically demonstrates how intrinsic molecular heterogeneity in the starting population is an important contributor to population diversity in immune responses, as has been demonstrated for specific loci (Wu et al, 2017). These molecular heterogeneities are often seen in high-dimensional data, but their significance has heretofore been unclear. Although we used B cells and cytokines as a model, the cell-intrinsic effects described herein operate in other scenarios, such as cancer drug resistance (Goyal et al, 2021 Preprint). To systematically understand these clonal effects, we characterized the transcriptomic similarity of clones. We detected CCGs in our in vitro samples, as well as in plasma cells isolated directly from human bone marrow. The set of CCGs was heavily enriched for immune response genes such as JCHAIN, MS4A1, and AIDCA, while simultaneously containing hundreds of genes which are likely relevant features of B-cell biology, but not yet investigated. Taken together, our data show how intrinsic clonal transcriptional identities are faithfully propagated during cell differentiation in vitro and in vivo. Our multimodal molecular measurements were crucial for characterizing these effects, and should allow more useful probabilistic models of the immune system (Hodgkin, 2018). These models of the immune system will benefit from continued systematic efforts to compile molecular information about cell states (Tabula Sapiens Consortium* et al, 2022).

Here we exploited the unique biology of B cells to gain insight into their differentiation processes. We conceptualized the BCR as an in vivo molecular recorder of antigen or germinal center experience. This allowed us to dissect the differences in activation programs between the progeny of memory and naive B cells. These differences are particularly interesting given the progeny would not be distinguishable via typical single-cell genomics workflows. We speculate similar intrinsic heterogeneity and cell fate biases exist in other immune cells (SanMiguel et al, 2020) but have yet to be fully described because of a lack of detailed clone tracking. In general, as lineage tracing and cell-recording technologies continue to develop, researchers will genetically record bespoke cell experiences, and use single-cell genomics to analyze their effects on intrinsic cell identities (Chen et al, 2022). Finally, we note that in vitro differentiation has moved out of research labs and into the clinic in the form of CAR-T therapies and regenerative medicine efforts. Single-cell genomics already offers powerful insight into cell therapeutics (Bode et al, 2021), but adding in vivo and in vitro lineage tracing technologies will yield critical additional power to characterize rare cells, improve targeting, and increase potency.

# Materials and Methods

### Sample collection

PBMCs were obtained from two healthy adult males aged 18 and 61 yr. LRS chamber product was diluted 1:4 in PBS + 2% FBS and PBMCs were isolated using a Ficoll gradient and Red Blood Cell Lysis. Cells were frozen in Cryostor CS10 according to the manufacturer's instructions. Human bone marrow aspirates from a healthy male aged 50–55 yr were obtained from AllCells. Mononuclear cells were isolated using Ficoll gradient and Red Blood Cell Lysis. Plasma cells from the bone marrow were isolated using StemCell EasySepEasySep Human CD138 Positive Selection Kit II.

### Experimental model and subject details

Study subjects gave informed consent, and protocols were approved by the Stanford Institutional Review Board and/or the AllCells Institutional Review Board.

### B-cell purification and culture

LRS chambers and Bone Marrow aspirates from healthy male donors were obtained. We used a Ficoll gradient separation (Stem Cell) and Red blood Cell lysis (BD Bioscience) to isolate mononuclear cells. PBMCs were frozen in CryoStor CS10. Bone marrow was obtained fresh from a fine needle aspirate of the iliac crest. Plasma cells were purified using the Stem Cell Cell EasySep Human CD138 Positive Selection Kit II and immediately loaded on the 10X Chromium Controller. For in vitro differentiation, PBMCs were thawed. B cells were purified by negative selection using the

StemCell B cell enrichment kit II. B cells were rested for 8 h in the B-cell medium (StemCell) at 37°C 5% $CO_2$ at a density of $2 \times 10^4$ cells per well. After 8 h, the Day 0 sample was taken and loaded on the 10X Chromium Controller. The remaining B cells were stimulated using the B-cell stimulation cocktail according to the manufacturer's instructions (Stem Cell). B-cell culture wells were thoroughly mixed every 24 h, to mitigate any spatial effects of the culture on particular B cells. On Day 4, the cells were split into three wells and restimulated. Every 24 h, these three wells were pooled, mixed, and redistributed into three new wells. On Day 8, the B cells were restimulated and separated into six wells, at which point the pooling and mixing was carried out using six wells.

### Methods note 1

The manufacturer of the product, Stem Cell Technologies, chooses not to disclose what is in their B Cell Expansion kit. We note that in spite of this major disadvantage, the product has two important benefits. First, this is the most robust and convenient of the multiple B-cell activation protocols we have used (i.e., worked on all donors and induced strong proliferation and differentiation in every experiment). Second, the contents of the stimulation cocktail are knowable and well-defined. These attributes are in contrast to almost all B-cell stimulation protocols of which are aware. Typical stimulation protocols use various types of feeder cells, cytokines, and FBS. In particular, the feeder cells and FBS, are highly variable between labs, lots, and even experiments. FBS has a myriad of issues (Gstraunthaler et al, 2013). With respect to B cells specifically, large differences in B cell–activation programs are observed between lots of FBS, as described here (Haniuda & Kitamura, 2019). In addition, we have detected various microbial nucleic acids in FBS, which may stimulate B cells (unpublished work).

We analyzed the contents of the kit provided by StemCell, as well as supernatants from the cell cultures, using the Luminex Human 80 plex (EMD Millipore) offered by the Stanford Human Immune Monitoring Center. We found the supplement mimics a strong T-cell engagement stimulus. It contains high amounts of CD40L, IL2, IL4, IL6, IL10, and IL21 (Fig S1A). The supplement also contains trace amounts of additional cytokines, potentially from the use of pooled human serum as a source of albumin in the medium. The supernatants of the cell cultures contain a number of other cytokines, which are likely to have been secreted by B cells or contaminant cells. At present we cannot determine whether there are microbial-derived molecules or small-molecules which mimic natural TLR agonists in the cocktail.

### Single-cell isolation and sequencing

On Day 0 and 4 B cells were washed two times in PBS + 1% BSA. On Day 8 and 12 live B cells were sorted then washed two times in PBS + 1% BSA. Then, cells were counted and loaded on the Chromium (10X Genomics) at a target loading of 15,000 cells per lane. Reverse transcription and cDNA amplification were performed using the Single Cell V(D)J kit V2 (10X Genomics). VDJ and gene expression libraries were prepared from each of the four time points using the Single Cell Library construction kit (V2). All library preparations were carried out according to the manufacturer's instructions, except for the use of custom constant region primers (Horns et al, 2016) for VDJ enrichment. On days

8, and 12, cells from culture wells were sorted by propidium iodide exclusion into cold PBS + 2% BSA, before proceeding to antibody staining. Libraries were sequenced using the Illumina Novaseq platform with paired-end reads of 26 and 98 bp.

### Preprocessing of single-cell sequencing data

We used Snakemake (Mölder et al, 2021) to manage the computational workflow. We used CellRanger 7 to map, count, and assemble reads from the sequencing libraries. We used the Immcantation docker (Gupta et al, 2015) pipeline and scirpy (Sturm et al, 2020) to reprocess the contigs assembled by CellRanger, distinguish bonafide single cells from multiplets, and assign clonal barcodes, which agreed with our in-house pipeline (Croote et al, 2018). We defined clones as cells with identical heavy-chain CDR3 regions and using the same IGHV gene. Single B cells were identified by the presence of a single productive heavy chain and a single productive light chain, yielding a total of 29,703 single B cells for analysis. All other cells were excluded from further analysis.

### Single-cell sequencing data analysis

All the code implementing all analyses is available on Github (https://github.com/michael-swift/seqclone). We generally describe the analysis procedures here. Gene expression analysis of single cells was performed using scanpy (Wolf et al, 2018), and exploratory analysis of the immune receptors with scirpy. Briefly, single-cell transcriptomes were log-transformed and normalized to counts per $10^4$ UMIs. We annotated cell types with the Celltypist algorithm using majority voting. Celltypist performed similarly to the annotation we created using known marker genes and standard clustering methods. Celltypist classified a plurality of B cells isolated from PBMCs as "Age-Associated B cells." The genes which the model used to label "Age-associated B cells" did not appear specific for this group of cells. Thus, we replaced that label with "B cells" and proceeded with our analysis. In all cases where DEGs were identified, we used the Wilcoxon-rank sum test adjusted for multiple testing using the Benjamini–Hochberg procedure as implemented in scanpy.

### Analysis of mutated versus germline outcomes

To explore the cell fate biases of memory versus naive progenitors, we binned VDJ sequences into heavily mutated, mutated, and germline categories as shown in Fig S2A. We used multiple different levels of reasonable cutoffs for these categories and found they did not change our general conclusions. We calculated the ratio of germline over mutated by dividing the number of germline cells by the number of mutated or heavily mutated cells in a given type of category, such as which IGH constant region is associated with the cell's VDJ or which Celltypist label is associated with a VDJ mutation state.

### Analysis of clonal fate bias

In general, we used the resampling and permutation methods to estimate confidence intervals and $P$-values for all clonal effects. For calculating the amount of fates available to a B cell, we performed Leiden clustering on the nearest neighbor graph, with a resolution of 1.2. We then permuted clonal labels within groups defined by the 10X

lane and mutation status. This accounted for possible batch effects, as well as removing the powerful effect of mutation status on cell fate, which allowed a more meaningful "progenitor cell to progenitor cell" comparison of fate bias. Our general conclusions of cell-fate restriction within clones were generally insensitive to different parameter choice. For the analysis of CSR, we specified a null model where the isotype of any given cell was randomly sampled from the representative population of isotype identities seen in Fig S3D. We then compared those models to the measured isotype identities associated with each cell in a clone to calculate a deviation from expectation. For identifying CCGs, we implemented the approach lucidly described here (Horton et al, 2018). Briefly, for each gene detected in more than five B cells, we averaged the SD of UMI counts within clonal groups. We compared this value to the same statistic, but where the clone labels of cells within each 10X lane are permuted. We performed that test 10,000 times to derive a null-distribution for the statistic which we used to generate Benjamini–Hochberg–corrected q-values. For calculating the clonal index, we divided the difference in the statistic between true labels and average permuted labels by mean, log-transformed UMI count for that gene in the whole population.

### Analysis of CCG enrichment

We calculated the overlap coefficient between size-matched sets of the DEGs, CCGs, and B-cell transcriptome genes. The set of DEGs was constructed from 400 of the top genes identified by differential expression analysis of the Celltypist defined clusters. The set of CCGs was constructed as the top 400 most significant CCGs. The sets of B-cell transcriptome genes were 400 randomly sampled genes from the B-cell transcriptome. The B-cell transcriptome was defined as genes expressed by more than 5% of B cells. Our conclusions of enrichment were insensitive specific choice of parameters (e.g., 400 genes versus 1,000 genes) or by the definition of B-cell transcriptomes.

### Luminex-EMD Millipore human 80 plex kits

This assay was performed by the Human Immune Monitoring Center at Stanford University Immunoassay Team. Kits were purchased from EMD Millipore Corporation and run according to the manufacturer's recommendations with modifications described as follows: H80 kits include three panels: Panel 1 is Milliplex HCYTA-60K-PX48. Panel 2 is Milliplex HCP2MAG-62K-PX23. Panel 3 includes the Milliplex HSP1MAG-63K-06 and HADCYMAG-61K-03 (resistin, leptin, and HGF) to generate a 9 plex. The assay setup followed the recommended protocol: Briefly, samples were diluted threefold (Panel 1&2) and 10-fold for Panel 3. 25 μl of the diluted sample was mixed with antibody-linked magnetic beads in a 96-well plate and incubated overnight at 4°C with shaking. Cold and room temperature incubation steps were performed on an orbital shaker at 500–600 rpm. Plates were washed twice with a wash buffer in a BioTek ELx405 washer (BioTek Instruments). After 1-h incubation at room temperature with biotinylated detection antibody, streptavidin-PE was added for 30 min with shaking. Plates were washed as described above and PBS added to wells for reading in the Luminex FlexMap3D Instrument with a lower bound of 50 beads per sample per cytokine. Each sample was

measured with duplicate replicates. Custom Assay Chex control beads were purchased and added to all wells (Radix Bio-Solutions). Wells with a bead count <50 were flagged, and data with a bead count <20 were excluded.

## Data Availability

All code used to preprocess the sequencing data can be found on Github: https://github.com/michael-swift/seqclone. The raw sequencing data from this publication have been deposited to the SRA database https://www.ncbi.nlm.nih.gov/bioproject/PRJNA908079 and assigned the identifier PRJNA908079. Public 10X genomics data for PBMCs and BMMNCs are available from 10X Genomics. We used (https://www.10xgenomics.com/resources/datasets/20-k-human-pbm-cs-5-ht-v-2-0-2-high-6-1-0, https://www.10xgenomics.com/resources/datasets/10-k-human-pbm-cs-5-v-2-0-chromium-x-2-standard-6-1-0, https://www.10xgenomics.com/resources/datasets/10-k-bone-marrow-mononuclear-cells-bmmn-cs-5-v-2-0-2-standard-6-1-0). Pre-processed data (AnnData objects, AIRR tables, tables of CCGs) are available upon request.

## Supplementary Information

## Acknowledgements

We thank Ivana Cvijovic, Elizabeth Jerison, Derek Croote, Bali Pulendran, Dan Jarosz, and Daria-Mochly Rosen for useful discussions and helpful comments on the manuscript. This work was supported by the National Science Foundation Graduate Research Fellowship Program (to M Swift).

### Author Contributions

M Swift: conceptualization, data curation, software, formal analysis, validation, investigation, visualization, methodology, and writing—original draft, review, and editing.
F Horns: conceptualization, supervision, and writing—review and editing.
SR Quake: supervision, investigation, and writing—review and editing.

### Conflict of Interest Statement

The authors declare that they have no conflict of interest.

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
