## [Reviewer comments · Life Science Alliance]

Life Science Alliance

Lineage Tracing Reveals Fate Bias and Transcriptional Memory in Human B cells

Michael Swift, Felix Horns, and Stephen Quake

DOI: <https://doi.org/10.26508/lsa.202201792>

Corresponding author(s): Michael Swift, Stanford Medicine

Review Timeline:

Submission Date:	2022-10-31
Editorial Decision:	2022-11-25
Revision Received:	2022-12-15
Accepted:	2022-12-16

Transaction Report:

Please note that the manuscript was reviewed through Review Commons and these reports were taken into account in the decision-making process of the Life Science Alliance.

1. General Statements [optional]

The reviews are on balance an accurate, thoughtful, thorough assessment of the manuscript. We appreciate the careful engagement with the B cell differentiation aspect of our work. We identify 2 major critiques from the reviews:

1. The manuscript should make stronger connections with existing literature on *in-vitro* and *in-vivo* B cell differentiation.

We agree the manuscript should be revised to interact more holistically and carefully with relevant B cell differentiation research. In this respect, the reviewers both help by pointing to high-quality and relevant literature that will be discussed and cited.

2. The cytokine mixture we used on the B cells was not defined / described in the manuscript. This fact hinders the interpretation of the data because B cells will respond to diverse stimuli in quite different ways.

We agree this hinders interpretation of the data, and the reviewers bring up astute points about different types of stimuli (TD vs. TI vs. TLR vs. BCR). Unfortunately, the manufacturer of the product, Stem Cell Technologies, will not disclose exactly what is in the product. Given we are in strong agreement with the reviewers on this point, we analyzed the cytokine contents of the cocktail and our cell culture supernatants using a luminex cytokine panel. We present a discussion of our findings on this data in a supplementary note and figure. We acknowledge this analysis is non-exhaustive, because it does not include possible additions of non-cytokine stimulants. However, we maintain it adds much clarity to the interpretation of the data.

We note that the contents of the stimulation cocktail are knowable and well-defined. These attributes are in contrast to almost all B cell stimulation protocols of which are aware. Typical stimulation protocols use various types of feeder cells, cytokines, and FBS (Fetal Bovine Serum). In particular, the feeder cells and FBS, are highly variable between labs, lots, and even experiments. FBS has a myriad of issues which are described here (<https://www.ncbi.nlm.nih.gov/pmc/articles/PMC8349753/>). Major variability, from genomic to phenotypic, has been described in laboratory cell lines like the ones used as feeder cells. With respect to B cells specifically, large differences in B cell activation programs are observed between lots of FBS, as described here (<https://www.ncbi.nlm.nih.gov/pmc/articles/PMC7854248/#r5>). Additionally, we have observed the presence of bovine viruses and other contaminants in FBS (unpublished data). Thus, the stimulation protocol we used is reproducible and robust in ways generally unseen by us in B cell stimulation literature. In summary, we view this cocktail as useful in a similar way to how FBS is useful to biologists – a major difference being that this cocktail is better defined and controlled. We provide similar thoughts in our supplementary note.

A final general point we will make is about the significance of our work, which appears to be lost on Reviewer #1. Similar technical and conceptual advances by our lab have been cited 1000s of times. Thus, we think the impact of our scientific approach speaks for itself. We concur that many of our results confirm and expand on previous literature about B cells. We deliberately chose to make this novel technical and conceptual advance in the well-studied system of B cell differentiation. This allows us to integrate our findings with prior literature and helps validate the general approach. We thank Reviewer #1 for their scholarly service by independently verifying our findings are coherent with existing literature, and suggesting important literature to cite.

In response to the reviews, we have edited the manuscript to reference even more of the papers in the field which report similar findings. Thus, our concordance with prior literature should be viewed as a strength of the manuscript. It shows readers of the manuscript the conceptual framework we use here is valid and can generate similar insights in less well-studied systems. For example, future studies will use this approach developed in non-B cells, non-human immune systems, and developmental biology in general. In response to the reviewers critique, we modified the discussion of our work in multiple places to emphasize these points.

2. Description of the planned revisions

Insert here a point-by-point reply that explains what revisions, additional experimentations and analyses are planned to address the points raised by the referees.

1) Which B cell activation protocol was used? No information is provided in the main text or supplementary information. Yet, this information is key to fully understand many of the

conclusions of this work (e.g., ... memory B cells are intrinsically two-fold more persistent in vitro (2A)), which largely depend on the nature of the stimuli used in the in vitro B cell culture.

We used the B cell activation protocol developed by StemCell Technologies as described in our methods section. We agree the reader and scientific community would benefit from additional information about this cocktail. To this end, we added a discussion of the cocktail to the supplementary information. We also used a cytokine analysis panel to analyze the cocktail, which provided detailed although non-exhaustive information about what is in the cocktail.

2) It would be informative to use more than one B cell activation program, e.g., CD40L with or without a cytokine as well as CD40L vs. CpG-DNA. Authors make broad statements about B cell fates without discussing the impact of a given signal on a given B cell fate. For instance, do memory B cells follow the same differentiation program upon stimulation with CD40L, IL-4 or a combination of CD40L and IL-4? How about differences between a TD signaling program such as that provided by CD40L and IL-10 and TI signaling program such as that provided by CpG-DNA and IL-10?

This is a good point. We agree stimulation using a panel of different agents would be a worthwhile experiment. It stands as a goalpost for future studies. Currently, performing single cell RNA sequencing on so many samples is both beyond the scope of this manuscript and very resource intensive.

3) Page 3, first line: After low quality and non-B cells (Fig S1A & B). What does this statement mean? The sentence seems incomplete.

Thank you for catching this typo. It is now clarified in the manuscript that we removed these cells bioinformatically.

4) First, we noted that non-B cells present in the input rapidly became undetectable by day 4, which shows the specificity of the cytokines for B cell expansion. Which cytokines are we talking about? No detail is provided.

We now provide our analysis of the stimulation cocktail, in Supplementary note 1 and Supplementary figure 1A. We still believe it is an interesting observation that this cocktail specifically stimulates B cells because many cytokines are not specifically B cell division signals, there were some impurities in the input population, and many cytokines are produced by the cultured cells themselves.

5) Plasmablasts were not distinguished from plasma cells.

We agree this is an interesting and important distinction to make. We have now distinguished between these classes of cells.

6) Critically, we observed no appreciable evidence of hypermutation in vitro (S2C), consistent with prior literature (Bergthorsdottir et al. 2001). This statement is vague, misleading and likely inaccurate for the following reasons. (a) The B cell culture conditions used by the authors are completely unknown. (b) It was shown that SHM can be achieved under specific in vitro B cell culture conditions that include the presence of activated CD4+ T cells (PMID: 9052835; PMID: 10092799; PMID: 10878357; PMID: 12145648). Did authors try to recapitulate those culture conditions?

We see how this statement could be misunderstood. We only claim not to observe evidence of hypermutation in our specific culture conditions, which is important for the inferences we make. We added language to make this more clear.

We did not try to recapitulate the conditions in the references supplied by the reviewer. We note that these references use cell lines and not B cells. While there is immensely valuable work done on cell lines, they behave very differently from actual cells and these findings may not be relevant to our human B cells.

7) Some of the reported findings are repetitive of previously published results and provide no additional new information. For example:

a) "Interestingly, we found mutated B cells were far more likely to express genes involved in T cell interaction (2B), suggesting Memory B cells are intrinsically licensed to enter an inflammatory state which activates T cells".

This evidence is already published (PMID: 7535180 among many other published studies).

We will cite this paper, which is a landmark study. We don't claim we are the first to discover a propensity of mutated B cells to present help to T cells, but note that we were able to observe this fact via lineage tracing in a single experiment, which is a conceptual and technical advance. Additionally, we report an entire transcriptional module of genes which are upregulated in memory B cells vs. Naive B cells exposed to the same stimulus. This adds to the systematic understanding of the Memory B Cell activation program.

b) "Instead, Naive B cells were biased toward expressing lectins and CCR7, suggesting Naive B cells are intrinsically primed to home into the lymphatic system and germinal centers (2B)".

This evidence is already published (PMID: 9585422 among many other published studies).

While this is an interesting and important paper referenced by the reviewer, we are unable to find anything similar to our claim about naive B cells in the reference provided. The investigators do not discuss intrinsic differences between memory and naive subsets when responding to the same stimulus.

8) We quantified the *in vitro* dynamics of CSR through the lens of mutation status, which revealed strongly different fate biases between germline and mutated cells (2D). Most strikingly, B cells which switched to IGHE were almost exclusively derived from germline progenitors: the ratio of germline IGHE cells to mutated IGHE cells was (8-fold - inf, 95 % CI). Also this evidence is not novel (PMID: 34050324 among other published studies) and, again, must reflect the presence of specific culture conditions that remain completely undisclosed. This is incredibly confusing.

Thank you for providing this reference, we were not aware of this interesting study. These studies are quite different and complementary. Differences between these studies likely reflect the fact that their B cells are isolated from a niche, rather than generated *in-vitro*. Most of the tissue-resident cells in their study are quite mutated, and thus are not the Naive B cells we are making a claim about. In fact, despite their claim of low mutational load, these cells would fall into the “mutated” or even “heavily mutated” categories we defined in our paper. Cells with mutation levels of 5% are not thought to be Naive in any classification scheme. Our study showed that, *in vitro*, IGHE B cells effectively came exclusively from germline progenitors, their study shows no such result. The novelty of this finding was appreciated by reviewer 2.

9) Authors should mention that non-switched memory B cells include IgD^{low}IgM⁺CD27⁺ and IgD⁺IgM⁺CD27⁺ memory B cells. Some authors define these distinct memory B cell fractions as marginal zone (MZ) or MZ-like B cells (please, notice that splenic MZ B cells recirculate in humans) and IgM-only B cells, respectively (PMID: 28709802; PMID: 9028952; PMID: 10820234; PMID: 11158612; PMID: 26355154; PMID: 15191950; and PMID: 24733829 among many other published studies).

We appreciate these points. We attempted to classify our B cells within this taxonomy and found no such separation clearly exists in single-cell RNA-seq profiles. Instead, we opted to re-classify our data with a state-of-the-art algorithm called celltypist (DOI: [10.1126/science.abl5197](https://doi.org/10.1126/science.abl5197)) which harmonizes cell annotations across a growing number of single-cell RNA sequencing studies. While this classification system is not currently mutually exclusive / completely exhaustive, we believe using this system provides standardization and data availability that are key for sharing results. As single-cell RNA-seq and flow/mass cytometry harmonize their classification systems, anyone should be able to transfer their preferred classification scheme to the cells profiled here.

10) Thus, CSR from IGHM cells did not meaningfully contribute to the abundance of IGHA⁺ cells in the population. Also this conclusion may be misleading and/or inaccurate. Indeed, an efficient class switching to IgA requires the exposure of naïve B cells to the cytokine TGF-beta in addition to a robust TD (CD40L) or TI (CpG DNA or BAFF or APRIL) co-signal. Was TGF-beta present in this culture?

This is a good point about TGF-beta and switching to IgA. Here is a clear example of the novelty and power of our approach, as well as the benefits of using a well-characterized system such as B cell differentiation. Lineage tracing clarifies between two explanations

for why there are IgA cells in the output population. One explanation is that non-IgA B cells in the input switch to IgA, driven by TGF-beta. Another explanation is that IgA cells in the input expand modestly and account for IgA cells in the output. Lineage tracing offers clear evidence that the latter explanation is true. Following from this, our approach allows us to make a strong inference that TGF-beta is not present in the incompletely determined cytokine mixture. We are not sure how this conclusion may be misleading or inaccurate, as it is a clear and simple description of our data, not a claim about what factors are necessary for switching.

11) In contrast, we noted that many intraclonal class-switching events appeared to be directly from IGHM to IGHE. Explanations involving unobserved cells with intermediate isotypes notwithstanding, these data illustrate the relative ease with which B cells can switch directly to IGHE. It is very difficult to interpret this statement, as no information regarding the B cell-stimulating conditions used is provided. In addition, relevant literature is not quoted (e.g., PMID: 34050324).

We clarify our discussion here to claim the ease with which peripheral blood IGHM B cells switch to IGHE. Again, lineage tracing has allowed us to distinguish between two very different population-level phenomena. One explanation is that undetected IGHE+ progenitors in the input population expanded rapidly and account for the IGHE+ cells. Another explanation is that cells class-switch to IGHE. Our data are consistent with the latter. We note that this validates the conceptual use of lineage tracing to understand rapid population dynamics in immune responses and cell differentiation protocols. This is a strength of our manuscript. We appreciate the the reviewer has furnished relevant studies, which we will cite.

12) Our data for IGHE cells contrasts with in vivo data which show IgE B cells to be: (1) very rare, (2) apparently derived from sequential switching (e.g. from IgG1 to IgE) (Horns et al., 2016; Looney et al., 2016), and (3) often heavily hypermutated (Croote et al., 2018).

While this reviewer agrees with the first comment (switching to IgE is relatively rare in vivo, at least in healthy individuals), the other statements are quite inaccurate. Indeed, unmutated extrafollicular naïve B cells from tonsils and possibly other mucosal districts directly class switch from IgM to IgE in healthy individuals, thereby generating a low-affinity IgE repertoire. In principle, low-affinity IgE antibodies may protect against allergy by competing with high-affinity IgE specificities. In allergic patients, high-affinity IgE clones emerge from class-switched and hypermutated memory B cells that sequentially switch from IgG1 or IgA1 to IgE as a result of specific environmental conditions, including an altered skin barrier (PMID: 22249450; PMID: 30814336; PMID: 32139586).

Moreover, in contrast to what stated by authors, sequential IgG1/IgA1-to-IgE class switching mostly occurs in allergic patients but is less frequent in healthy individuals, where IgE specificities are less mutated (PMID: 30814336). Along the same lines, IgE is heavily mutated only in allergic individuals with significant molecular evidence of sequential IgG1/IgA1-to-IgE

class switching (PMID: 30814336; PMID: 32139586). Overall, the data provided by Swift M. et al. are largely confirmatory of previously published evidence.

We appreciate the clarification of this complex field and will cite the relevant literature. We also agree with the reviewers assessment that our data are validated by other approaches and groups. We see that our discussion of IgE B cells should have included that caveat that we are discussing IgE B cells detected in the peripheral blood. We have restricted claim to the suggestion that if our conditions mimic such niches where B cells switch to IgE, there are clearly efficient mechanisms which limit the amount of circulating IGHE B cells mechanisms in comparison to other isotypes.

Taken together, these data suggest that while direct switching to IGHE from Naive progenitors is trivial in vitro, niche factors or intrinsic death programs efficiently limit their generation or lifetime in vivo." I cannot understand this conclusion, which seems to contradict earlier statements.

We hope we have clarified via the above comment.

13) I am not sure I learned much regarding the "cell-intrinsic" fate bias and transcriptional memory of B cells after reading this elegantly presented but confusing and superficially discussed manuscript (please, see also comments 15-23).

We understand the reviewer is confused about various aspects of our manuscript and appreciate the opportunities to clarify. We show cells with broad identities (such as germline vs. mutated or naive vs. memory) respond differently to the same stimulus. These are cell intrinsic fate biases. We quantify them and provide statistical bounds on the effect sizes of these differences, which to our knowledge has not been done. We agree with the reviewer that in the case of memory and naive B cells, much is already known about their biases. We recapitulate some of this knowledge, while adding a quantitative and an unbiased transcriptomic lens with which to view the biases. However, our analysis moves beyond cell types broadly defined, and focuses on the concept that each clone is a cell state or identity, where some of the identity may be faithfully propagated over generations and other information may not be. To this end, we tracked the transcriptome of clones during differentiation. We show that B cell clones share highly similar cell fates, implicating cell-intrinsic heterogeneity as a major contribution to diversity in immune responses. We note this reviewer did not critique this aspect of our work. The review also did not critique Figure 3 or 4, in which we present a quantitative analysis of which transcriptional programs are maintained by B cells and contribute to their clonal identity. Finally, via our analysis of human long-lived plasma cells, we report these transcriptional identities are observed in-vivo, over long time scales. This type of cell-intrinsic bias has not been studied or described to our knowledge. These findings were of particular interest to reviewer 2 and other readers of the manuscript.

MINOR COMMENTS

Thank you for reading the manuscript carefully and providing these comments and observations. We have fixed all clerical errors that were pointed out. We also responded to some of these minor comments here, and made changes to the manuscript to clarify.

1) Figures 1B, S1C and S1D are not referred to in the text. x

2) 2B in the Text is 2D in the Figure. x

3) 2D in the Text is 2E in the Figure. x

4) Figure 2D seems to show only 10 genes. Please, clarify.

We clarify in the manuscript that we present the top differentially expressed genes

5) 2E in the text is 2F in the Figure. x

6) Figure S3B is not indicated in the text. x

7) Figure 3E is not indicated in the text. x

8) Figure S4A is not indicated in the text. x

9) In some sections of the text, Figure panels are not sequentially discussed, which makes the text very difficult to follow. x

Reviewer 2:

Major comments:

On p.3 the authors assume that a B cell with an unmutated BCR in the time course arose from a naive B cell progenitor. However, it is also possible that it arose from a IgM memory B cells since they also contain a non-negligible proportion of cells with 0 mutations. This was initially

seen already in the Klein, J Exp Med, 1998 paper and later confirmed by e.g. Weller et al, J Exp Med, 2008 and Wu et al, Front Immunol, 2011. And since the authors herein and others have demonstrated that IgM memory B cells have a high proliferative capacity it is possible that IgM memory B cells are overrepresented among those unmutated BCRs seen in the cultures.

The finding that IgM memory B cells are highly proliferative is not novel. It has been demonstrated by other groups before and one good example is Seifert et al, PNAS, 2015 where IgM memory B cells proliferated significantly more to BCR stimulation than naive or IgG memory B cells. However, it is also shown that IgG memory B cells are more responsive to TLR9 stimulation than IgM memory B cells as demonstrated by e.g. Marasco et al, Eur J Immunol, 2017. This is not discussed by the authors and should be added into the discussion for context of their finding by scRNAseq methods.

These are astute points. We incorporated a more nuanced discussion of the prior literature about highly proliferative IgM memory B cells, which have been reported before. We also added a figure which identifies the genes associated with proliferative clones in Figure 3d, which adds to our understanding of the gene regulatory networks which govern IgM memory B cell behavior. We appreciate the reference to the Seifert et al paper, which is relevant and high quality work. We concur that a discuss of Marasco would be helpful, especially because it is unknown if a TLR9 agonist is in the stimulation cocktail, but their data would suggest there is not.

The notion that a memory transcriptional program can be induced without SHM is not novel and this should be brought up in the discussion. One paper showing a memory transcriptional program in unmutated memory B cells is Kibler et al, Front Immunol, 2022.

We were not aware of this literature and have now cited it in our discussion of this finding.

The observation that memory B cells are more likely to enter an inflammatory state and support T cells has been suggested by other groups (Seifert et al, PNAS, 2015; Magri et al, Immunity, 2017, Grimsholm et al, Cell Reports, 2020).

We have now cited and discussed a number of papers which contain similar findings. We note that we add to the holistic understanding of this phenomenon via our single cell transcriptomic approach.

Please provide the age distribution of the peripheral blood samples as well.

We have now provided the age distribution of the peripheral blood samples

Please show flow cytometry analysis of the cultures to assist in assessing subset distribution, viability and plasma cell differentiation for each time point. This can be provided as supplementary information.

We did not use flow cytometry for subset distribution and measurements of differentiation per se, only to exclude non-viable cells and we have now made this clearer in the methods section. We also now include representative plots show our sorting strategy.

The stimulation cocktail used for this study, what does it contain? This needs to be specified in the manuscript and not only referring to the manufacturer. This has major impact on the results since different stimulatory agents will induce different pathways.

This is a valid point that we addressed in our response to reviewer 1. See supplementary note 1 and Figure S1A for our analysis of the stimulation cocktail.

Minor comments:

Please avoid the term plasma B cells, does it refer to plasmablasts and/or plasma cells?

Thank you for the suggestion, we have modified our language to refer to plasmablasts and plasma cells separately.

November 25, 2022

RE: Life Science Alliance Manuscript #LSA-2022-01792-T

Michael Swift
Stanford University
94301

Dear Dr. Swift,

Thank you for submitting your revised manuscript entitled "Lineage Tracing Reveals Fate Bias and Transcriptional Memory in Human B cells". We would be happy to publish your paper in Life Science Alliance pending final revisions necessary to meet our formatting guidelines.

- please address the remaining Reviewer 1 and 2' s comments
- please upload your main manuscript text as an editable doc file
- please upload your main and supplementary figures as single files and add a separate section for your figure legends to the main manuscript text
- please add a Running Title, alternate abstract/summary blurb and a category for your manuscript to our system
- please add an ORCID ID for the secondary corresponding author-they should have received instructions on how to do so
- please add the Twitter handle of your host institute/organization as well as your own or/and one of the authors in our system
- please add the Author Contributions and a conflict of interest statement to the main manuscript text
- please use the [10 author names, et al.] format in your references (i.e. limit the author names to the first 10)
- please add figure callouts for all of your figures to your main manuscript text; please note that if you add a figure callout for a specific panel (Figure 1A), then you need to add figure callouts for every panel in that figure
- please include the section Supplementary Material note 1 and 2 into your main Materials and Methods section
- please add a separate Data Availability section at the end of Materials and Methods section in which you include your deposited analysis code

A. FINAL FILES:

B. MANUSCRIPT ORGANIZATION AND FORMATTING:

Sincerely,

Reviewer #1 (Comments to the Authors (Required)):

The reviewers have answered fully to my concerns that I had previously. I congratulate the authors to the revised version that have removed my main concern about the content of the stimulation cocktail. It now makes it easier to interpret the data for the reader. A minor comment: please check the entire manuscript for typos because I found a few. Also please avoid capital letters on naive and memory B cells in sentences.

Referee cross comments: I fully agree with reviewer 2 that proliferative germinal centre B cells will most likely not be induced by the cocktail in this study and the lack of BCR stimulation will make it very unlikely to induce anything close to a germinal center B cell. Please rephrase this as suggested by reviewer 2.

Reviewer #2 (Comments to the Authors (Required)):

This is a revised version of a nicely written and elegantly presented work that dissects the contribution of intrinsic cell states to cell differentiation fates. Authors identified molecular programs that may explain how cell-intrinsic fate biases influence human B cell differentiation. This is a novel and important contribution.

Compared to the original manuscript, the revised manuscript has considerably improved and is now much clearer. In addition to adding relevant references and correcting a series of quite distracting clerical errors, authors partially disclosed the composition of the commercially available cocktail used to stimulate B cells. In the revised manuscript, Figure 1a and Supplementary Note 1 indicate that the stimuli included in this cocktail included CD40L, IL-2, IL-4, IL-6, IL-10 and IL-21, which support the differentiation data presented in subsequent figures. For the sake of clarity, I would suggest to disclose these stimuli also in the first paragraph of the Results.

That being said, I doubt that the above stimuli are sufficient to induce the large cluster of "proliferative germinal center B cells" shown in Figure 1d. Indeed, germinal center B cells do not recirculate and I cannot recall earlier works showing their differentiation from circulating B cells. It would be reassuring to validate the germinal center nature of the above B cell cluster by showing nuclear Bcl-6 expression, which is a hallmark of germinal center B cells. Without this evidence, it may be difficult to

convince readers that this cluster includes bona fide germinal center B cells. I suggest authors to consider this point carefully.

Reviewer #1 (Comments to the Authors (Required)):

The reviewers have answered fully to my concerns that I had previously. I congratulate the authors to the revised version that have removed my main concern about the content of the stimulation cocktail. It now makes it easier to interpret the data for the reader. A minor comment: please check the entire manuscript for typos because I found a few. Also please avoid capital letters on naive and memory B cells in sentences.

Referee cross comments: I fully agree with reviewer 2 that proliferative germinal centre B cells will most likely not be induced by the cocktail in this study and the lack of BCR stimulation will make it very unlikely to induce anything close to a germinal center B cell. Please rephrase this as suggested by reviewer 2.

Reviewer #2 (Comments to the Authors (Required)):

This is a revised version of a nicely written and elegantly presented work that dissects the contribution of intrinsic cell states to cell differentiation fates. Authors identified molecular programs that may explain how cell-intrinsic fate biases influence human B cell differentiation. This is a novel and important contribution.

Compared to the original manuscript, the revised manuscript has considerably improved and is now much clearer. In addition to adding relevant references and correcting a series of quite distracting clerical errors, authors partially disclosed the composition of the commercially available cocktail used to stimulate B cells. In the revised manuscript, Figure 1a and Supplementary Note 1 indicate that the stimuli included in this cocktail included CD40L, IL-2, IL-4, IL-6, IL-10 and IL-21, which support the differentiation data presented in subsequent figures. For the sake of clarity, I would suggest to disclose these stimuli also in the first paragraph of the Results.

That being said, I doubt that the above stimuli are sufficient to induce the large cluster of "proliferative germinal center B cells" shown in Figure 1d. Indeed, germinal center B cells do not recirculate and I cannot recall earlier works showing their differentiation from circulating B cells. It would be reassuring to validate the germinal center nature of the above B cell cluster by showing nuclear Bcl-6 expression, which is a hallmark of germinal center B cells. Without this evidence, it may be difficult to convince readers that this cluster includes bona fide germinal center B cells. I suggest authors to consider this point carefully.

We appreciate the additional feedback from reviewers.

- **All instances of Naive and Memory B cells are now consistent with the reviewers preference.**

- **We now disclose the identity of the cytokines used in the main text.**
- **With respect to the “proliferative germinal center B cells”, we respectfully decline to change the text and figures. We used a data-dependent algorithm called Celltypist to annotate our cells. We believe this is conceptually the ideal way to classify cells. We note in the main text that this is a data-dependent, automatic annotation via an algorithm.**

December 16, 2022

RE: Life Science Alliance Manuscript #LSA-2022-01792-TR

Mr. Michael Swift
Stanford Medicine
94301

Dear Dr. Swift,

Thank you for submitting your Research Article entitled "Lineage Tracing Reveals Fate Bias and Transcriptional Memory in Human B cells". It is a pleasure to let you know that your manuscript is now accepted for publication in Life Science Alliance. Congratulations on this interesting work.

DISTRIBUTION OF MATERIALS:

Again, congratulations on a very nice paper. I hope you found the review process to be constructive and are pleased with how the manuscript was handled editorially. We look forward to future exciting submissions from your lab.

Sincerely,
